Journal of Data-centric Machine Learning Research (2025)      Submitted 03/25; Revised 09/25; Published 09/25

# A Model Zoo on Phase Transitions in Neural Networks

**Konstantin Schürholt**                                          KONSTANTIN.SCHUERHOLT@UNISG.CH
*Department of Computer Science*
*University of St. Gallen, Switzerland*

**Léo Meynent**                                                        LEO.MEYNENT@UNISG.CH
*Department of Computer Science*
*University of St. Gallen, Switzerland*

**Yefan Zhou**                                                    YEFAN.ZHOU.GR@DARTMOUTH.EDU
*Department of Computer Science*
*Dartmouth College, USA*

**Haiquan Lu**                                                        HAIQUANLU@U.NUS.EDU
*Department of Electrical and Computer Engineering*
*National University of Singapore, Singapore*

**Yaoqing Yang**                                                 YAOQING.YANG@DARTMOUTH.EDU
*Department of Computer Science*
*Dartmouth College, USA*

**Damian Borth**                                                    DAMIAN.BORTH@UNISG.CH
*Department of Computer Science*
*University of St. Gallen, Switzerland*

**Reviewed on OpenReview:** *https: // openreview. net/ forum? id= zJRWvNpdIr*

**Editor:** Hongyang R. Zhang

## Abstract

Using the weights of trained Neural Network (NN) models as data modality has recently gained traction as a research field — dubbed *Weight Space Learning* (WSL). Multiple recent works propose WSL methods to analyze models, evaluate methods, or synthesize weights. Weight space learning methods require populations of trained models as datasets for development and evaluation. However, existing collections of models — called 'model zoos' — are unstructured or follow a rudimentary definition of diversity. In parallel, work rooted in statistical physics has identified phases and phase transitions in NN models. Models are homogeneous within the same phase but qualitatively differ from one phase to another. We combine the idea of 'model zoos' with phase information to create a controlled notion of diversity in populations. We introduce 12 large-scale zoos that systematically cover known phases and vary over model architecture, size, and datasets. These datasets cover different modalities, such as computer vision, natural language processing, and scientific ML. For every model, we compute loss landscape metrics and validate full coverage of the phases. With this dataset, we provide the community with a resource with a wide range of potential applications for WSL and beyond. Evidence suggests the loss landscape phase plays a role in applications such as model training, analysis, or sparsification. We demonstrate this in an exploratory study of the downstream methods like transfer learning or model weights averaging.

**Keywords:**   Model Zoo, Weight Space Learning, Neural Networks, Phase Transition

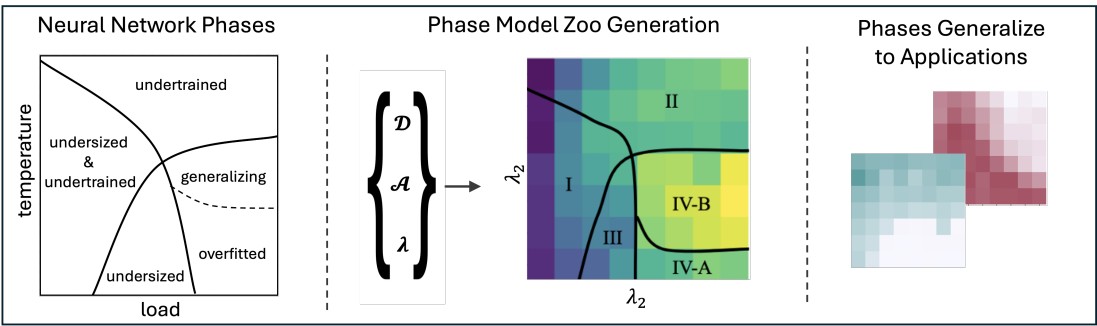

Figure 1: Overview of the approach. **Left:** Phases of Neural Networks with qualitatively different properties have been identified (Martin and Mahoney, 2019a; Yang et al., 2021). **Middle:** We propose to use phases as a quantifiable diversity metric and generate datasets of trained models by varying their datasets $\mathcal{D}$, architectures $\mathcal{A}$ and training parameters $\lambda$ s.t. they cover all known phases across different domains. **Right:** Phases manifest beyond evaluation metrics in applications of pre-trained models, e.g., in weight averaging. Our Phase Model Zoos allow systematic evaluation of methods that use pre-trained models.

## 1 Introduction

Training NNs has become standard practice on many tasks. However, there are aspects of model training as well as the trained models that are still not fully understood. How to parameterize training and what model size to use remains a question for researchers and practitioners. With models growing in size, the computing power required to train them also increases exponentially (OpenAI, 2024). It is therefore becoming more and more common to reuse pre-trained models from open model repositories like Hugging Face (HuggingFace, 2025) and fine-tune them for more specific tasks. However, the choice of a specific pre-trained model for fine-tuning or transfer learning strategies is non-trivial.

**Weight Space Learning and Model Analysis.** There is growing research interest in addressing these gaps by analyzing collections of trained models. Using neural networks as data modality – dubbed *weight space learning* (WSL) has become its own research field[1]. Within the field of WSL, populations of models are used in a variety of domains: Applications include predicting how well models generalize (Jiang et al., 2020), or detecting poisoned models (IARPA, 2024). Other methods use WSL torwards Neural Architecture Search to identify the relation between weights and model performance (Martin et al., 2021; Zhou et al., 2024; Liu et al., 2024). Different work re-uses existing trained weights to extend conventional transfer learning strategies (Yosinski et al., 2014; Mensink et al., 2021) by combining multiple models, via a combination of their outputs (Polikar, 2012; Ganaie et al., 2022; Mohammed and Kora, 2023), knowledge distillation (Liu et al., 2019; Shu et al., 2021), or by directly combining their weights (Wortsman et al., 2022a,b; Mitchell et al., 2022; Ainsworth et al., 2022; Ilharco et al., 2022; Rame et al., 2023; Ramé et al., 2024; Navon et al., 2024), to build combined models with improved in- and out-of-distribution performance.

---

1. ICLR 2025 hosts the first workshop on Neural Networks as Data Modality `https://weight-space-learning.github.io/`

There is also a recent interest in using model populations to infer model properties from weights or activations (Yak et al., 2018; Jiang et al., 2020; Eilertsen et al., 2020; Martin et al., 2021). Closely related, there is research on learning representations of NNs, for either model analysis (Schürholt et al., 2021; Ashkenazi et al., 2022; Navon et al., 2023; Zhou et al., 2023a; De Luigi et al., 2023; Lim et al., 2023; Andreis et al., 2023; Herrmann et al., 2024; Navon et al., 2024) or model generation (Ha et al., 2017; Zhang et al., 2019; Knyazev et al., 2021; Schürholt et al., 2022a; Peebles et al., 2022; Knyazev et al., 2023; Zhang et al., 2023; Shamsian et al., 2024; Schürholt et al., 2024; Putterman et al., 2024; Wang et al., 2024; Soro et al., 2025).

**Phase Transitions of NNs.** Another line of research investigates phase transitions in NNs and connects them to loss landscape metrics (Martin and Mahoney, 2019a; Yang et al., 2021). Motivated by statistical physics, they identify two main types of hyperparameters in NN training: the noisiness of the training process, dubbed temperature, and the amount of data relative to the size of the model, dubbed relative load. Using that notion, they identify distinct phases with qualitatively different model properties on the temperature-load landscape (Yang et al., 2021). Phase transitions have been studied extensively in the machine learning literature (Schwarze et al., 1992; Seung et al., 1992; Martin and Mahoney, 2019a,b). While phase transitions are typically studied theoretically under certain limits, e.g., an infinitely wide NN (Lewkowycz et al., 2020), we observe qualitatively similar properties in models with a practical size. Within phases, models are relatively homogeneous, with abrupt changes from one phase to the next. Along the load dimension, one example of a phase transition is *double descent* (Nakkiran et al., 2019). Neural scaling laws describe the load and temperature relation, e.g., as power laws to train models to specific phases (Hestness et al., 2017; Rosenfeld et al., 2020; Gordon et al., 2021; Hoffmann et al., 2022; Sorscher et al., 2022).

**Exisiting populations of Neural Networks.** The field of *weight space learning* uses models as data, and thus requires datasets of trained models. Most freely accessible models are part of large public repositories like Hugging Face (HuggingFace, 2025) or the PyTorch model hub (You, 2018), opening up new opportunities such as transfer learning (Prottasha et al., 2022; Lyu et al., 2021) and benchmarking (Chiang et al., 2024; Zheng et al., 2024). Within those collections, however, models are of varying quality and mostly unstructured. The hyperparameters, data, and training strategy are often documented rudimentary, which makes them unsuitable for research on model populations. Researchers studying model populations sometimes publish the models they used for their work — but the design of these populations and the available information are usually catered to their specific goal.

To fill that gap, structured populations have been published as model zoo datasets, in domains as varied as computer vision (Unterthiner et al., 2020; Eilertsen et al., 2020; Schürholt et al., 2022b; Falk et al., 2025a), adversarial robustness assessment (Croce et al., 2020), bioimaging (Ouyang et al., 2022) or Earth observation (Honegger et al., 2023b,a). These populations, however, contain only models trained on one domain, and consider the diversity of the model in their populations only in terms of their generating factors. They report the distribution in model performance and model similarity, but it remains unclear whether that indicates conceptually different models, or whether they are variations of similar representations.

**Our WSL dataset covers multiple phases.**    To address that gap in datasets for *weight space learning*, we combine the concept of model zoos with loss landscape and phase information (Yang et al., 2021) and augment the model zoo blueprint with a new notion of diversity. Instead of aiming for diversity in model performance, which may or may not contain qualitatively different models, we create the model zoos with different architectures, model sizes, tasks and domains such that all the phases on the loss landscape are covered. This allows us to quantitatively show that the distinct, qualitatively different phases of models exist in our model zoos and generalize across architectures and domains.

Going beyond the models at regular training stage, our work shows the existence of phases in downstream methods such as fine-tuning, transfer learning, pruning, ensembling, etc. Such methods are usually evaluated on one or a few models, without explicit consideration for their phase diversity. Without that context, it is hard to attribute performance gains or the lack thereof to the method or the phase of the pre-trained model. Model zoos that cover all phases could help systematically evaluate methods that rely on pre-trained models and identify where they work. This, in turn, may not only provide performance and robustness information but also a good signal to guide further research in this area.

In summary, with this dataset, we make the following contributions:

- We propose a new notion of diversity in model zoos by covering different model phases that relate to qualitatively different model properties using loss landscape taxonomy.

- We systematically create 12 model zoos that include all phases. The zoos cover computer vision, natural language and scientific machine learning (SciML) models, contain different architectures of different sizes and are trained on various datasets. It contains a total of ~2.5k unique neural network models between 11K and 900M parameters and more than 60k checkpoints.

- We annotate the models with performance and loss landscape metrics, and include checkpoints for multiple epochs. We quantitatively validate that our zoos are diverse and cover known phases.

- We discuss the benefit of our dataset for the ML community: as quantifiable diverse model dataset for weight space learning, and contributing to better and more systematic evaluation of methods that use pre-trained models.

- We make our dataset available at `https://phasetransitions.modelzoos.cc`

In the following, we first introduce metrics to taxonomize phases in loss landscapes (Sec. 2). Subsequenlty, we detail the generation of the model zoos (Sec. 3). We then evaluate that phases exist in our model zoos, generalize across different zoos, across domains, architectures, tasks and model sizes (Sec. 4). Finally, we discuss concrete applications in weight space learning for our model zoos (Sec. 5).

## 2 Loss Landscape Taxonomy

**Phases in Neural Networks Loss Landscapes.**    The motivation for introducing phases and phase transitions in NN loss landscapes is rooted in statistical mechanics, where such phenomena explain qualitative changes in system behavior (Martin and Mahoney, 2019a). Phases represent distinct regions in the parameter space where the system's properties are

homogeneous or change smoothly, while phase transitions mark abrupt changes in these properties. In NNs, phases manifest in terms of generalization performance. A prominent example for such a phase transition is the double descent pheonmenon (Nakkiran et al., 2019), a phase transition along the axis of model size (Belkin et al., 2019; Liao et al., 2020; Derezinski et al., 2020). Similar empirical observations have been made recently on the *emergent* abilities of large language models (Wei et al., 2022), in which non-smooth transitions can occur when some training hyperparameters (such as the model size) are modified. However, it is not conclusive whether these emergent abilities are indeed sharp phase transitions or merely due to specific ways of experimental measurements (Schaeffer et al., 2024). These phases and transitions are expected due to the complex, high-dimensional nature of NN optimization, where varying control parameters like data noise and training iterations can lead to qualitatively different properties, akin to physical systems undergoing phase changes. Motivated by statistical physics, Martin and Mahoney (2019a) identify two main types of hyperparameters in NN training: the noisiness of the training process, dubbed temperature, and the amount of data relative to the size of the model, dubbed relative load. Using that notion, distinct phases with qualitatively different model properties on the temperature-load landscape can be identified (Yang et al., 2021). Interestingly, the phases and phase transitions can be linked to the structure of the loss landscape (Yang et al., 2021), especially its global structure. We note that the study of global loss landscapes has been an active area of research, and it is generally understood that the properties of neural networks cannot be fully captured by local sharpness alone (Fort and Jastrzebski, 2019; Fort et al., 2020). Yang et al. (2021) present the first empirical attempt to quantify the transition from a globally well-connected to a globally less well-connected loss landscape. Specifically, metrics such as the *training* loss, the sharpness of local minima, and mode connectivity or representation similarity computed on the training data can be used to identify the phase of a model.

**Loss Landscape Metrics.** Yang et al. (2021) categorize phases in load-temperature variations using four metrics. The first metric is the training loss, which evaluates whether the training data is interpolated. The other metrics describe the sharpness of the local minima, the similarity between models trained using different random seeds, and the connectivity between different local minima of the loss landscape. It should be noted that Yang et al. (2021) used a certain set of metrics to measure these loss landscape properties, but there are alternative metrics available. For example, the sharpness of local minima can be measured using *adaptive sharpness metrics* (Andriushchenko et al., 2023; Kwon et al., 2021), while similarity can be measured using *disagreement* (Theisen et al., 2023).

We define the loss landscape metrics following Yang et al. (2021). Let $\boldsymbol{\theta} \in \mathbb{R}^m$ denote the learnable weight parameter, and let $\mathcal{L}$ be the loss function. We compute metrics using the train set unless stated otherwise.

**Hessian-based Metrics.** The Hessian matrix $\mathbf{H}$ at a given point $\boldsymbol{\theta}$ can be represented as $\nabla^2_{\boldsymbol{\theta}} \mathcal{L}(\boldsymbol{\theta}) \in \mathbb{R}^{m \times m}$. The largest eigenvalue $\lambda_{\max}(\mathbf{H})$ and trace $\mathrm{Tr}(\mathbf{H})$ are used to summarize the local curvature properties in a single value. Specifically, a larger value of the top eigenvalue or trace indicates greater sharpness.

**Mode Connectivity.** The mode connectivity assesses the presence of low-loss paths between different local minima and reflects how well different solutions are connected in the parameter space, indicating smoother and more generalizable loss landscapes. It is common to fit Bézier curves $(\gamma_\phi(t)$ between two models $\boldsymbol{\theta}$ and $\boldsymbol{\theta}'$, and subsequently compute mode connectivity mc as

$$\texttt{mc}(\boldsymbol{\theta}, \boldsymbol{\theta}') = \frac{1}{2}(\mathcal{L}(\boldsymbol{\theta}) + \mathcal{L}(\boldsymbol{\theta}')) - \mathcal{L}(\gamma_\phi(t^*)),$$

where $t^* = \underset{t}{\mathrm{argmin}} \left| \frac{1}{2}(\mathcal{L}(\boldsymbol{\theta}) + \mathcal{L}(\boldsymbol{\theta}')) - \mathcal{L}(\gamma_\phi(t)) \right|$. Here, mc $< 0$ indicates a loss barrier between the two models and hence poor connectivity. mc $> 0$ reveals lower loss regions between the models which indicates poor training. mc $\approx 0$ indicates well-connected models.

**CKA Similarity.** Centered Kernel Alignment (CKA) (Kornblith et al., 2019) is used to evaluate the similarity between representations learned by different NNs, providing a measure of consistency and robustness in feature learning. CKA helps to understand how similar the learned representations are across different minima, linking representation similarity to landscape structure and generalization performance. The CKA between output logits $\mathbf{X}$ and $\mathbf{Y}$ generated by $\boldsymbol{\theta}$ and $\boldsymbol{\theta}'$ is computed as

$$\texttt{cka} = \frac{\mathrm{HSIC}(\mathbf{K}, \mathbf{L})}{\sqrt{\mathrm{HSIC}(\mathbf{K}, \mathbf{K}) \cdot \mathrm{HSIC}(\mathbf{L}, \mathbf{L})}}$$

where HSIC is the Hilbert-Schmidt Independence Criterion and $\mathbf{K}$ and $\mathbf{L}$ are the Gram matrices of $\mathbf{X}$ and $\mathbf{Y}$, respectively.

**Phase Taxonomy.** Based on loss landscape metrics, the NN hyperparameter space is divided into five distinct phases, as depicted in Figure 2.

- **Phase I (underfitted & undersized)**: Train loss is high; Hessian metrics are relatively large (indicated by a rugged basin); Mode connectivity is negative (indicated by a barrier between two local minima).
- **Phase II (underfitted)**: Train loss is high; Hessian metrics are relatively large; Mode connectivity is positive.
- **Phase III (undersized)**: Train loss is small; Hessian metrics are relatively small (smooth basin); Mode connectivity is negative.
- **Phase IV-A (overfitted)**: Train loss is small; Hessian metrics are relatively small; Mode connectivity is near-zero; CKA similarity is relatively small.
- **Phase IV-B (generalizing)**: Train loss is small; Hessian metrics are relatively small; Mode connectivity is near-zero (no barrier between minima); CKA similarity is relatively large.

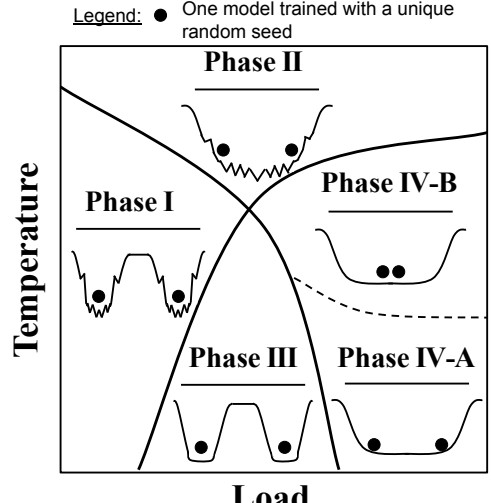

Figure 2: Five-phase taxonomy in NN hyperparameter space (Yang et al., 2021), varied by load-like and temperature-like parameters. Our zoos cover all five phases.

Yang et al. (2021) use these metrics, along with fixed thresholds, to partition training into distinct phases. Training loss separates early phases (Phase I and II) from later ones (Phase III and IV). Mode connectivity, depending on whether it is positive or negative, differentiates Phase I from Phase II. Hessian trace and CKA similarity are then used to further divide Phase IV into two sub-phases. Zhou et al. (2024) formalize this procedure as a hierarchical decision tree, where thresholds for each metric can be learned using a small number of reference models. This optimization is performed using the bounded Brent method to maximize prediction accuracy on 'failure modes' of models. They show that the resulting phase boundaries can be transferred effectively to new tasks or model architectures.

## 3 Phase Transition Model Zoos

To create a population of models that covers relevant phases and can be used to evaluate for phase transitions, we train structured populations of NNs with several architectures on different datasets following the blueprint introduced by Unterthiner et al. (2020) and Schürholt et al. (2022b). Within each *model zoo* population, we systematically vary load-like and temperature-like hyperparameters to realize all of the phases. For every model in the zoo, our dataset includes multiple checkpoints (i.e. saved model weights), at different training epochs. We annotate these samples with performance metrics (training and test loss and accuracy), as well as the loss landscape metrics outlined in Section 2. Since the exact localization of phase boundaries is still a topic for ongoing research, we annotate each model with its loss landscape metrics as quantitative ground truth, from which different methods can derive slightly different phase transition boundaries. We further track loss and accuracy on train and — if available — validation and test sets. In the following, we first detail our model zoo generation scheme, which we use to create 12 zoos on three domains: computer vision, natural language and physical systems. Further details can be found in Appendix A.2. Subsequently, we analyze our models with conventional performance metrics, but also with loss landscape metrics to quantify the qualitative diversity of the proposed zoos and validate that all of the phases are realized.

**Computer Vision Model Zoos.** Ten computer vision zoos form the foundation of our dataset and demonstrate the generalization of the phase transitions accross architectures and datasets, while the language and SciML zoos show generalization of the phase transitions to other domains and tasks. We generate the computer vision zoos from combinations between two architectures {ResNet, ViT} of different sizes and four standard computer vision datasets {SVHN, CIFAR-10, CIFAR-100, TinyImagenet} (Netzer et al., 2011; Krizhevsky and Hinton, 2009; Le and Yang, 2015). Details on the model zoo configurations can be found in Table 4 in the Appendix. We choose ResNet (He et al., 2016) and ViT (Dosovitskiy et al., 2021) architectures for the zoos because of their proliferation in computer vision to achieve representative populations. Importantly, ResNet and ViT architectures allow smooth scaling of model width and thus model capacity for the same architecture without the need to adjust the learning scheme. Similarly, the set of datasets evaluate generalization accross different data distributions. Taken together, the vision zoos contain $\sim 1.8$k unique models and $\sim 55$k checkpoints.

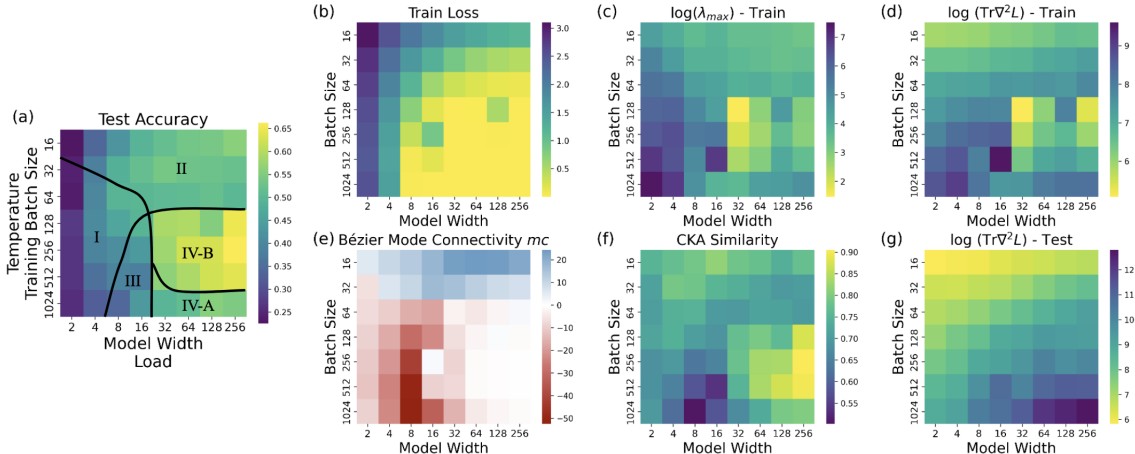

Figure 3: Performance and loss landscape metrics for the CIFAR-100 ResNet-18 model zoo. **(a):** test accuracy and phases of models in the zoo. **(b):** training loss; **(c-g)** different loss landscape metrics introduced in Section 2. Our model zoos cover all phases identified in previous work (Yang et al., 2021).

**Large Language Model Zoo.** For the natural language zoo, we train GPT-2 (Radford et al., 2019) models, following the reference implementation in (Karpathy, 2025). We use decoder-transformer GPT-2 to include coverage of modern LLMs in our dataset, which have since largely converged to decoder-only transformers. The GPT-2 architecture can be scaled just like the ViTs by changing the width, allowing smooth variations of the model capacity. We train on openwebtext which replicates the original dataset of GPT-2 (Gokaslan and Cohen, 2019). We use Chinchilla scaling laws to determine the reference ratio of a GPT-2 nano model to data and train on 2.29B tokens (Hoffmann et al., 2022). Notably, work on scaling laws established relations between model size and data amount for compute-optimal performance. This suggests that phase distributions identified on GPT-2 nano-sized models generalize to much larger models if the data amount is scaled accordingly, within the range of the scaling laws. This significantly increases the expressiveness of the zoo. Interestingly, this is exactly what the relative load parameter as the ratio of model capacity to data suggests, and what makes the phase layout on these abstract axes powerful. In its final grid, the language zoo contains 264 unique GPT-2 models with $\sim 1.3$k checkpoints.

**SciML Model Zoo.** Lastly, we also create zoos of models for the simulation of physical systems. With that, we want to evaluate whether phases and phase transitions generalize even beyond the conventional deep-learning domains. Specifically, we train physics-informed neural networks (PINNs) (Raissi et al., 2019) to learn the solution to partial differential equations, following the implementation setup in (Krishnapriyan et al., 2021). To curate the dataset, we simulate the 1D convection partial differential equation (pde) and randomly sample the domain's collocation points (position, time) as the data samples. Each data sample is comprised of position and time as input and the PDE solution value as the label. The training objective includes minimizing the loss of data prediction, boundary/initial conditions, and physics-based regularization. We train a 4-layer fully connected NN with 50 neurons per layer and tangent activation function using the L-BFGS optimizer. By varying temperature and load paramters, we train a zoo with 700 unique PINNs and 7 000 checkpoints.

**Load and Temperature Variations.** For all model zoos on all domains, we introduce specific variations in the training hyperparameters to obtain models in all phases. Previous work identifies the phases on the surface spanned by load-like and temperature-like hyperparameters (Martin and Mahoney, 2019a; Yang et al., 2021) The load-like parameters can be understood as the quantity and/or quality of data relative to the model capacity. Temperature represents the noisiness of the training process. Following previous work, we realize variations in load by changing the model width. Increasing the model width increases model capacity and thus decreases the relative load. By varying the width, we achieve variations in model capacity without changing the architecture or having to adapt the training scheme. In ResNets, the width directly refers to the number of channels. In transformer models (ViTs and GPT-2) , we realize width by changing the `model_dim` parameter, i.e. the size of intermediate representations. In SciML models, we change the load by varying the PDE convection coefficient, as it changes the complexity of the data. To realize variations in temperature, we choose to adapt the batch size or learning rate. Here, lower batch size and higher learning rate increase the noisiness of the training updates and this increases the temperature. For every combination on the grid, we train three different models using three random seeds. All other hyperparameters are kept constant between the models.

## 4 Empirical Evaluation of Phases in the Model Zoos

The model zoos are designed to cover different phases. In the following, we validate phase coverage by testing for the phases introduced by Yang et al. (2021) summarized in Section 2. Full phase plots for all 12 zoos and further details can be found in Appendix A.3.

### 4.1 Phases Systematically Emerge Across all Zoos

Our experiments demonstrate that phase transitions are consistently present in the training of neural networks across all domains, architectures, and datasets evaluated. The exact phase layout is affected by the architecture, dataset, and data augmentation strategies. The specific characteristics of these phases remain consistent with the four-phase taxonomy outlined in previous studies, validating our experiment setup.

A central reason we focus on load-like and temperature-like parameters is that they unify the effects of many hyperparameters into two axes. Empirically, models occupying the same phase on these axes exhibit qualitatively similar properties—regardless of which specific training factors (e.g., batch size, learning rate, or data size) gave rise to those load and temperature values. Consequently, rather than exhaustively exploring every hyperparameter combination, we can vary these two axes to capture all known phases. This is not only supported by prior work—where phase boundaries remain stable across different load–temperature choices (Appendix D in Yang et al. (2021))—but also reflected in our results, suggesting strong generality of phase transitions to a broad range of architectures and training regimens.

**Phases Generalize Computer Vision Architectures, Model Sizes and Datasets.** As illustrated in Figure 3 and in Figures 6-17, the phases manifest clearly in the combination of loss landscape metrics such as Hessian trace, mode connectivity, and CKA similarity. While different methods to locate phase transitions may identify slightly differ boundaries,

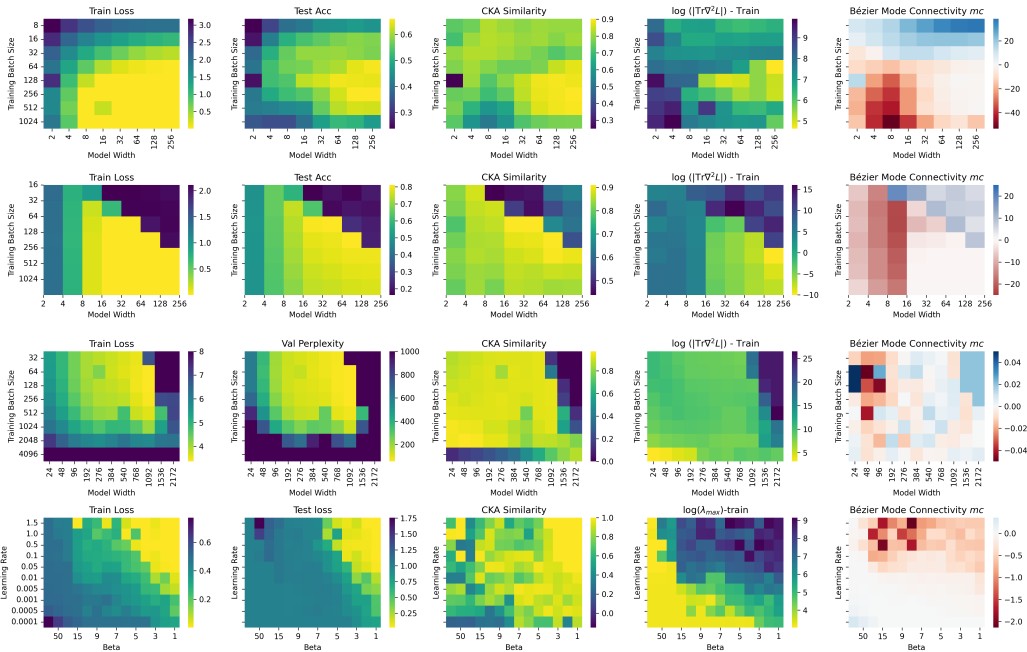

Figure 4: Phase plots accross different architectures, domains and tasks. Phases and phase transitions emerge in train loss, test performance, and loss landscape metrics, across different domains, tasks, and architectures. The exact layout is determined by the architecture, domain, and task. For example, transformers (ViT and GPT-2) have a sharp transition in the high-temperature low-load (top right) regime, which ResNets and PINNs do not show.

the evaluation of the performance and loss landscape metrics in the phase plots clearly demonstrates the coverage of the phases described in previous work. In particular, Phase IV-B, associated with the best test accuracy, is marked by low loss and high generalization performance. On the ResNet zoos, our results reveal that learning rate decay plays a significant role in shaping the phase distribution. Specifically, decaying the learning rate by $1e4$ under cosine annealing increases the area of Phase IV (well-trained regime) while reducing the presence of Phase II (under-trained regime), as the effect of batch size variations diminishes. This may be an indication of why learning rate decay is so successful. Our experiments show that the phase transitions generalize across different datasets, architectures, and training regimes. ViTs, for instance, display a sharper transition from Phase II to IV when trained without heavy augmentation (Figure 8). GPT-2 models exhibit a comparable pattern (Figure 16), suggesting that phase layouts remain consistent across distinct architectures, including transformers.

**Phases Generalize Across Domains.** Notably, the phases and phase transitions on the load-temperature landscape emerge on vastly different domains, from computer vision and classification to natural language and generation, and even physics and simulation, see Figure 4 and Figures 6-17 in the Appendix. While the exact layout varies between the zoos, and is affected by architecture and task, our results suggest that the fundamental drivers of phases translate directly. Accross all domains and architectures, the high test performance region is contained in the region with low train loss. However, the low train loss region is

significantly larger. As described in previous work, loss landscape metrics can be used to identify the layout of best test performance. The combination of high CKA similarity, low curvature (hessian trace) and low mode connectivity error computed on the training data predict high test performance. There are two exceptions that bear further investigation: First, in SciML models, while the phase layout aligns with previous work (Geniesse et al., 2024), the high performance phase has high hessian traces. The rason for that may lie in the training problem of PINNs, which is known to have particular properties (Cheng and Na, 2024; Krishnapriyan et al., 2021). In our zoos, low learning rates may cause PINNs to get trapped in flat local minima, while global minima are sharp. Second, mode connectivity on the GPT-2 models is computed on the loss rather than on the prediction error which adds noise, due to the high dimensionality of the classification problem.

These experiments suggest that since the phases generalize to different domains and architectures, findings of specific phases in specific applications on one domain can also be found in all domains. While some phase transitions on non-CV domains have been described before, the direct match is to the best of our knowledge novel and underscores the expressiveness of phases and its usefulness for the composition of diverse zoos. By confirming the presence of all known phases, we establish that this dataset is not only a tool for studying phase transitions but also a resource for designing and testing phase-aware training algorithms. Next, we turn to downstream methods to illustrate why phases matter post-training.

## 4.2 Phase Transitions in Neural Network Methods

Building on the observation that phase coverage describes performance and generalization during training, we now examine how phases influence four popular downstream methods: fine-tuning, pruning, ensembling, and weight averaging. In practice, many widely-used methods rely on *pre-trained* models—raising the question of whether and how a model's phase continues to matter when its weights are reused. Below, we evaluate four canonical techniques and discuss how our phase-covering model zoos enable a more comprehensive analysis of their properties. Across all these downstream applications, we find distinct phases in downstream performance, see Figure 5. Some of them overlap with the (pre-)training phases, some are distinct. These results are notable for three reasons: **(i)** it demonstrates that phase transitions exist broadly **(ii)** the load-temperature axis of our zoos is general enough to cover phase transitions in several downstream applications, beyond pre-training, significantly improving the applicability of our zoos; **(iii)** the datasets facilitate research of *where* these methods work or fail beyond point-wise evaluation, and build towards phase-aware method design.

**Fine-Tuning and Transfer Learning.** When fine-tuning, e.g. a CIFAR10-pretrained ResNet-18 on STL-10 (Figures 6,12, bottom left), we find that networks ending in a well-trained (Phase IV) regime yield stronger adaptation. Conversely, under-fitted or over-fitted phases struggle to reach comparable accuracy despite identical fine-tuning settings. This aligns with earlier observations that good phases not only benefit the original task but also confer more adaptable representations. For less aligned tasks like Tiny-Imagenet to STL-10 (Figure 13), we notice a phase shift from pre-trained to fine-tuned models. Understanding the relation between pre-training tasks and phase alignment using our model zoos can help pre-train or choose models to fine-tune in a more targeted fashion.

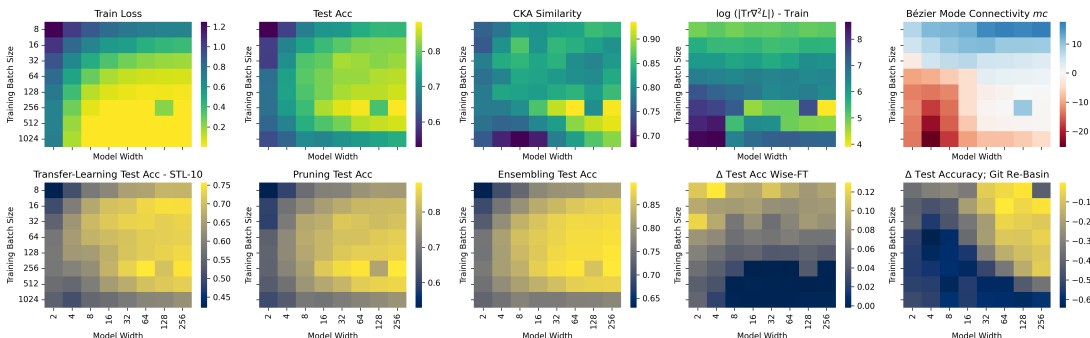

Figure 5: Phases and phase transitions of ResNet-18 models trained on CIFAR-10 in **top:** pretraining and loss-landscape metrics and **bottom:** downstream methods. Phase transitions emerge in downstream methods like fine-tuning, pruning and weight averaging. We use a different color palette for the downstream methods figures in order to make them easily identifiable.

**Pruning.** Pruning has been shown to exhibit phase-dependent outcomes (Zhou et al., 2023b). We confirm that Phase IV networks often preserve accuracy better under high sparsity, while certain under-fitted/low-connectivity phases degrade quickly (Figure 5 bottom, second from left). By comparing across load-temperature grids, our model zoos help pinpoint where and why pruning works best.

**Ensembling.** Combining model predictions (ensembling) has been investigated for phase transitions (Theisen et al., 2023). Our experiments confirm that ensembles tend to improve performance in load-temperature regions where local minima are well-connected (Figure 5 and 8). Notably, the phase of high-performance ensembles is larger than the individual model high-performance Phase IV, demonstrating the robustness benefit of ensembling with respect to pretraining settings. In practice, that corresponds to phases with near-zero mode connectivity barriers. Where mode connectivity is negative, ensembling provides less benefit. The zoo's phase annotations thus indicate not just whether ensembling helps, but *which* configurations best exploit ensemble diversity.

**Weight Averaging.** Averaging weights has recently gained popularity to improve robustness or combine knowledge (Izmailov et al., 2019; Wortsman et al., 2022a; Ilharco et al., 2022). We test averaging across training epochs (Wise-FT (Wortsman et al., 2022b)) and across pre-training seed (git re-basin (Ainsworth et al., 2022)). Our experiments (Figures 6–15) reveal distinct phase-dependent outcomes: Wise-FT is notably effective in high-temperature regions (Phases I or II) where averaging reduces noisy training updates, while in flatter phases (e.g., IV) it can yield little additional benefit. Git Re-Basin, on the other hand, hinges on positive mode connectivity; negative-connectivity phases (I or III) remain difficult to merge without performance loss. Hence, Hessian trace and mode connectivity strongly govern weight-averaging success, underscoring the value of our phase-annotated model zoos for pinpointing when and where averaging is most fruitful. Here in particular, point-evaluations of merging a few models lack the big picture necessary to build understanding.

## 5 Model Zoo Applications in Weight Space Learning

The "weight space" perspective treats trained NN weights as a data modality in its own right, enabling various novel applications such as model analysis, merging, editing, generation etc. Most of these approaches require a training set of model weights, as well as structured, diverse evaluation sets. Our phase-annotated model zoos directly cater to this need: they provide large, systematically varied populations of trained models with a quantifiable notion of diversity, alongside key loss-landscape metrics. They can be used to train weight-space models and therefore can facilitate the development of future methods in the area. They can also be used as a diverse, structured, standardized evaluation set for such methods. In the following, we highlight recent developments in weight space learning and illustrate how our dataset can contribute to research in those areas.

### 5.1 Learning Dynamics Analysis

In this paper, we focus on the phase transition phenomenon as studied in the statistical physics literature (Engel, 2001). This line of work links physics models, such as the Curie-Weiss model of magnetization, to problems in statistical inference. It often uses strong analytical tools like the replica method to compute free energy and characterize the phase transitions of learning systems. For a comprehensive overview, see Zdeborová and Krzakala (2016). At the same time, there is a growing body of research on phase transitions in learning dynamics. These studies show that the behavior of learning dynamics/trajectories can shift sharply as time evolves or as hyperparameters change. Some of this work also connects to statistical physics, while others develop independently. For example, Lewkowycz et al. (2020) identify three learning phases under different initial learning rates: lazy, catapult, and divergent. They show that a large but non-divergent learning rate yields the best test performance. As another example, Baity-Jesi et al. (2018) describe three stages of training: exploration of the loss landscape, aging dynamics with an increasing number of flat directions, and a final phase resembling diffusion. Extending our analysis to capture more of these phenomena in various learning dynamics is an important direction for future work.

### 5.2 Model Training

A core challenge in deep learning is hyperparameter tuning, which often relies solely on validation performance (Jaderberg et al., 2017; Li et al., 2020). By shifting toward a weight space perspective, practitioners can leverage loss-landscape signals—e.g., Hessian trace, mode connectivity—to guide models into more favorable phases (e.g., flatter minima). Recent works demonstrate that such information can enable faster or more reliable optimization (Yao et al., 2018; Zhou et al., 2023b, 2024). Lack of availability of specialized model zoos with loss landscape metrics however limit their evaluation to few model zoos. Our model zoos address this limitation, as they contain checkpoints covering model trainings in length, and are enriched with corresponding loss landscape metrics. With our broad coverage of the load-temperature axes, as well as multiple data modalities, our dataset offers opportunities to study the influence of these parameters on the training of the models in a variety of contexts and test methods across different domains and architectures.

Table 1: Evaluating the performance of phase-aware hyperparameters optimization approaches compared to a random search baseline, on ResNet-18 models. We report average over 50 runs as well as standard deviation. For all three model zoos evaluated, phase-aware hyperparameters optimization outperforms the baseline.

| Algorithm | CIFAR-10 | CIFAR-100 | TinyImageNet |
|---|---|---|---|
| **Random Search** | $0.012 \pm 0.006$ | $0.026 \pm 0.009$ | $0.037 \pm 0.012$ |
| **Phase-Aware Optimization** | $\mathbf{0.038 \pm 0.004}$ | $\mathbf{0.072 \pm 0.002}$ | $\mathbf{0.116 \pm 0.007}$ |

We demonstrate the potential of our dataset for phase-aware training algorithms with a simple hyper-parameter optimization experiment to maximize performance gains in a single step. We consider any state on the grid, and attempt to change temperature or load like parameters to improve performance. That is, for each configuration, we perform one-step hyperparameter optimization (to any other load-temperature cell) and measure the resulting performance improvement. As stand-in for validation-based methods, we use a random search that keeps the post-hoc better cell in validation metrics. We compare against a simple decision rule based on the current loss landscape metrics.

**Random Search Baseline**: For each model configuration, we randomly select one of three tuning actions: increase model width, increase batch size, or decrease batch size. The magnitude of adjustment is also randomly sampled from a predefined range 1-5.

**Loss Landscape-Guided Search**: For each model configuration, we first classify its phase using four loss landscape metrics and the phase boundaries by identifying thresholds similarly to Yang et al. (2021). We then apply deterministic tuning directions:

- Phase I and III: Increase model width

- Phase II: Increase batch size

- Phase IV-A: Decrease batch size

- Phase IV-B: No parameter adjustment

While tuning directions are deterministic based on phase classification, tuning magnitudes are randomly sampled to maintain a fair comparison with the baseline. Please note that this one-step procedure can easily be extended to multi step optimization. We focus on single step solely for evaluation purposes. To ensure statistical reliability, we repeat each experiment 50 times and report both mean performance gains and standard deviations. We report results in Table 1.

The results clearly show that using loss landscape information to optimize model hyper-parameters significantly improves performance and efficiency of a single optimization step. We would further like to refer the reader to Zhou et al. (2024), who have systematically evaluated the directional benefit of loss landscape metrics for hyper-parameter optimization over conventional validation metrics.

### 5.3 Model Analysis

Weight spaces can also be leveraged to infer model properties — test accuracy (Eilertsen et al., 2020; Unterthiner et al., 2020), generalization power (Schürholt et al., 2022a), backdoor presence (Langosco et al., 2023), task (Herrmann et al., 2024), INR class (De Luigi et al., 2023; Navon et al., 2023), relations to other models (Horwitz et al., 2024) etc. — directly from the model weights. Most of these works train their own model zoos, limiting the possibility to effectively compare their performance, and making the evaluation on a wide variety of training parameters and modalities prohitively expensive.

Our dataset offers a broad distribution of architectures, training states, and documented loss-landscape metrics, creating an ideal environment for developing or benchmarking weight space predictors, in particular to predict tasks, hyperparameters, performance and loss landscape metrics. Because our zoos deliberately include "bad" phases (I, III) as well as "good" phases (IV), methods that learn from this data will be more robust across real-world scenarios. We demonstrate the usability of our dataset by conducting a small pilot experiment following Unterthiner et al. (2020): we extract simple weight statistics (e.g. per-layer means and quintiles) for each model, then use a linear probe to predict both performance and loss-landscape metrics. Table 2 reports the $R^2$ values on a held-out test split, suggesting that curvature (log Hessian) and representation similarity (CKA) are partly learnable from raw weights. Even mode connectivity (MC)—which depends on pairs of models—shows signs of predictability, albeit less strongly.

These preliminary findings underscore two key insights. First, the metrics that define each *phase* appear sufficiently structured that simple regression models can partially recover them from the weights alone. Second, this partial predictability could serve as an efficient *guidance signal* during training or hyperparameter opti-

Table 2: Predicting ResNet-18 performance and loss-landscape properties from raw weight statistics: $R^2$ on unseen test sets.

| Data | Test Acc | GGap | CKA | $\log(\mathbf{Tr}\nabla^2 L)$ | MC |
|---|---|---|---|---|---|
| SVHN | 0.92 | 0.67 | 0.41 | 0.48 | 0.68 |
| CIFAR10 | 0.54 | 0.53 | 0.46 | 0.61 | 0.81 |
| CIFAR100 | 0.89 | 0.67 | 0.13 | 0.79 | -2.93 |
| TinyImgNet | 0.90 | 0.63 | 0.28 | 0.72 | 0.52 |

mization, instead of computing expensive Hessian or connectivity measures. While more advanced predictors will be needed to achieve high fidelity, our results highlight how phase-annotated model zoos open the door for practical methods that leverage *weight-space learning* to navigate the loss landscape.

### 5.4 Model Editing

Working on the weights of NNs as a modality opens up opportunities to directly alter them. Existing applications include re-aligning models with regard to permutation symmetries (Ainsworth et al., 2022; Navon et al., 2024), domain adaptation (Navon et al., 2023), knowledge unlearning (Meng et al., 2023), or model merging (Ilharco et al., 2022). Here again, the wide variety of models in our zoos along hyperparameters, architectures and modalities will allow easy development and standardized evaluation of such methods using phase information signals.

## 5.5 Model Generation

Finally, there is a growing push to *generate* NN weights, either to initialize new models without full retraining or to combine multiple networks for improved performance (Schürholt et al., 2022a; Peebles et al., 2022; Schürholt et al., 2024; Wang et al., 2024; Putterman et al., 2024; Soro et al., 2025; Meynent et al., 2025; Falk et al., 2025b). However, as we argue above, existing datasets to that end either contain only well-trained models or do not control for diversity in a quantifiable way. Our dataset provides precisely the sort of phase-diverse, architecture-diverse examples that generative models can learn from. By conditioning on quantifiable quality metrics that describe the phase, or on loss landscape information like curvature or connectivity, a generator trained on our zoos could be trained to produce custom weights with predictable properties.

## 6 Discussion

**Limitations**   While our model zoos cover the domains of vision, language and science, most Phase Transition Model Zoos are composed of classification models in the computer vision domain. We chose to explore phase transitions in vision exhaustively, and to demonstrate the generalization of phases as a concept with the additional zoos in language and science. Our work presents itself as a first step to making model zoos comprehensively cover phase transitions for a variety of applications, and we leave its extension to further tasks and domains for future work.

**Conclusion**   The Phase Transition Model Zoos represent the largest structured collection of models across architectures, tasks and domains annotated with detailed loss landscape metrics. With it, we provide the research community with a powerful dataset for weight space learning, and a tool to evaluate neural network performance across different phases. By systematically covering phase transitions, it allows the study of robustness, generalization, and failure modes of deep learning methods in a much more nuanced, comprehensive, and reliable way.We demonstrate the relevance of phase transitions by identifying phases in methods like fine-tuning, transfer learning, pruning, ensembling, and weight averaging. We show that these phases significantly affect performance and that their impact varies from one method to another, indicating valuable insights that can be gained with our dataset beyond conventional performance metrics.

## Broader Impact Statement

Our work introduces a dataset of neural networks that systematically spans multiple phases of training across different architectures and domains. By providing controlled diversity in the weight space, we aim to accelerate research on weight space learning, a growing field that treats trained weights as a data modality. This dataset can help researchers analyze when and how different training phases arise, develop phase-aware training and downstream methods, and gain deeper insights into the fundamental properties of neural networks.

This project inherits the typical risks of large-scale machine learning research and progress of the field. While we hope our dataset can save ressources in phase aware research, the computational resources for creating, analyzing, and extending such model zoos can be significant.

## Acknowledgments and Disclosure of Funding

KS, LM and DB are supported by the Swiss National Science Foundation research project grant 10001118 and SPRIND through the ModelFoundry project. YZ and YY are supported by DOE under Award Number DE-SC0025584 and funding from Dartmouth College.

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

## Appendix A. Dataset documentation

Please find the dataset at `https://phasetransitions.modelzoos.cc`[2]. The repository contains instructions on how to access the dataset, code to use it, code used to generate the model zoos and the loss landscape metrics, as well as other related information.

### A.1 Model Zoo Contents

In the main paper, we described the generation of the model zoos as well as explored their performance and phase information. Here, we detail the contents of the datasets. A model zoo contains a set of trained Neural Network models. For each of the zoos, we fix architecture and task combinations and introduce variations in temperature-like and load-like parameters. We realize temperature variations by varying the batch-size, and load variations by varying the model width. We chose the training parameters and variation range such that the phases and phase transitions described by Yang et. al (Yang et al., 2021) can be observed. We repeat each temperature-load combination 3 times with different random seeds to compute loss landscape metrics and get robust results.

For every model sample, there are model state checkpoints at intervals throughout training. The checkpoints are in PyTorch format, which uses pickle to save ordered dicts. We will provide code to convert the checkpoints to framework-neutral file formats. We annotate these samples with performance metrics (training and test loss and accuracy), as well as the loss landscape metrics (hessian eigenvalues, Bézier mode connectivity, CKA similarity). We add additional results like model averaging performance, where applicable to individual models. The model zoos are generated with `ray.tune`[3] and largely follow their experiment structure. Each model in a population is contained in one folder. Checkpoints are kept in subfolders for the corresponding epochs. Each model is annotated with a `config.json` file to re-create the model exactly. Performance metrics are tracked for every epoch and saved in a `results.json` file for every model. For a subset of epochs, we add loss-landscape metrics. All model zoos contain full meta-data configs and self-contained Pytorch code, s.t. they can be re-instantiated exactly, re-trained, or fine-tuned. All code to train grids, evaluate, compute loss landscape metrics and model averaging is available alongside the data. Further, we provide code to i) recreate the model zoo datasets, ii) compute loss-landscape metrics, iii) load the models, and iv) re-create the figures in the main paper. In order to allow easy use of the dataset, we plan to make adequate PyTorch dataset classes available upon publication.

This section will be updated upon dataset publication. Indeed, several statements are intentionally left vague as of now. Our dataset is large and will require a careful choice of what to include to balance the dataset's utility with its size. This will influence, in particular, the number of checkpoints that we include per model.

### A.2 Model Zoo Generation

We generated the dataset for common computer vision, language, and SciML tasks and architectures to show generalization and maximize applicability to the community. We fixed the load-temperature grids by exploring the boundary cases first and establishing the

---

2. Back-up link: `https://github.com/ModelZoos/PhaseTransitionModelZoo`
3. `https://docs.ray.io/en/latest/tune/index.html`

Table 3: Summary of the content of each model zoo included in our dataset.

| Architecture | Dataset | # models | Load-like param. | Temperature-like param. |
|---|---|---|---|---|
| ResNet-18 | SVHN | 192 | Model width | Batch size |
| ResNet-18 | CIFAR-10 | 192 | Model width | Batch size |
| ResNet-18 | CIFAR-100 | 192 | Model width | Batch size |
| ResNet-18 | TinyImagenet | 192 | Model width | Batch size |
| ResNet-50 | SVHN | 192 | Model width | Batch size |
| ResNet-50 | CIFAR-10 | 192 | Model width | Batch size |
| ResNet-50 | CIFAR-100 | 192 | Model width | Batch size |
| ResNet-50 | TinyImagenet | 192 | Model width | Batch size |
| ViT | CIFAR-10 | 147 | Model width | Batch size |
| ViT | CIFAR-100 | 147 | Model width | Batch size |
| GPT-2 | OpenWebText | 264 | `model_dim` | Batch size |
| MLP (PINN) | 1D Convection | 700 | `beta` | Learning rate |

presence of phase transitions, then filling in more resolution. For the vision and language zoos, we chose batch size as a temperature-like parameter, as it's a common hyperparameter to vary. For the load-like parameter, we vary the width of the models by changing the number of channels or the size of the internal token, respectively. For the SciML zoo, we adapt the scheme to the specifics of the domain. To vary the temperature, we chose to change the learning rate to keep the batch size and collocation points constant. To vary the load, we change the complexity of the domain by varying $\beta$ parameter of the physical system. We base the GPT-model implementation and training setup on NanoGPT Karpathy (2025) and use Chinchilla scaling laws to find the ideal number of training steps Hoffmann et al. (2022). The full list of model zoo hyperparameters is given in Tables 4,5 and 6. For vision zoos, we use Random Cropping, horizontal flipping, and random rotations. Training ViTs on CIFAR100 required stronger data augmentation to achieve competitive performance. Therefore, we have applied a combination of random cropping, random erasing, color jitter, and RandAugment (Cubuk et al., 2019). After the initial tuning of the grids, the training of the model zoos was done on 16 DGX H100 GPUS in 30 days. The computation of loss landscape metrics was performed on the same hardware in 14 days.

## A.3 Model zoo evaluation

In this section, we test the general validity of the trained models as representatives of real-world models in a structured dataset. An overview of the models at the end of training is given in Tables 7 and 8. The results confirm that models are trained to competitive performance for their respective sizes.More nuanced information on the distribution of model performance on the temperature-load grid is shown in Figures 6 through 16. Similar to previous work, the zoos show distinct low train-loss regions, with smaller embedded regions within that generalize well. Test performance generally improves with decreasing load (increasing width), with a distinct peak phase where temperature and load are low enough, but not too low. The generalization gap correspondingly shows a superposition of both patterns. Further applications or loss landscape metrics likewise show clear phase transitions. Remarkably, the phase layout and loss landscape metrics generalize across different domains, tasks, architectures, and datasets.

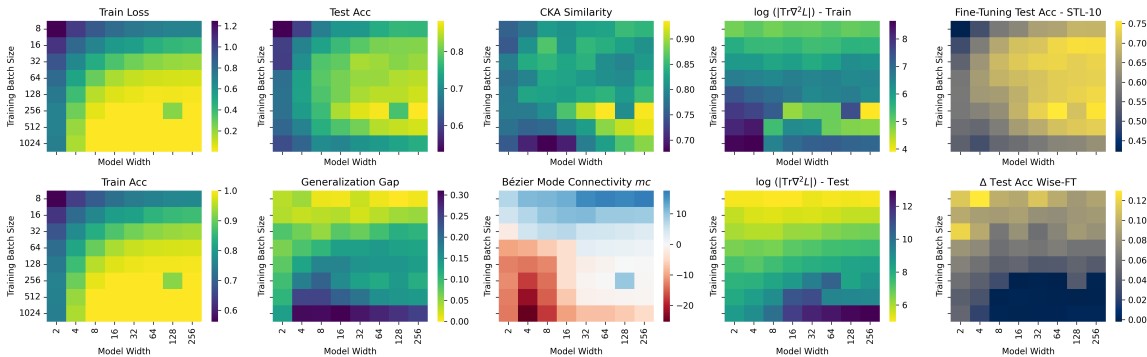

Figure 6: Phase plots for the CIFAR-10 ResNet-18 model zoo, showing distinct phase transitions in performance and loss-landscape metrics, fine-tuninng, and weight averaging.

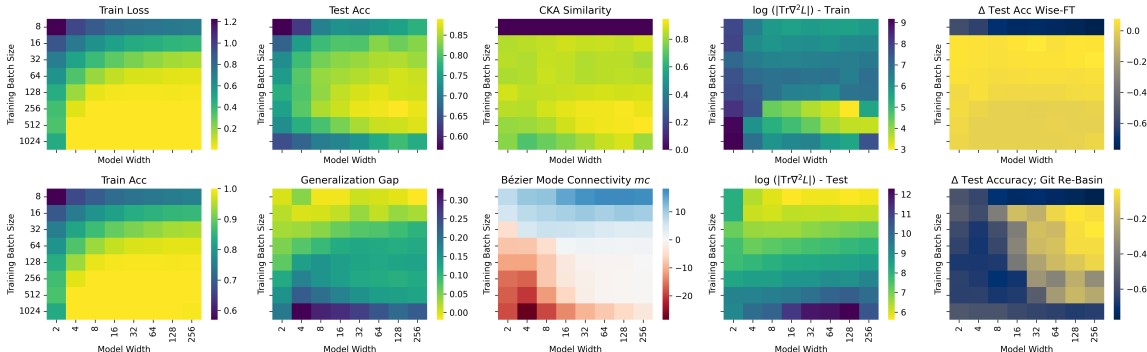

Figure 7: Phase plots for the CIFAR-10 ResNet-50 model zoo, showing distinct phase transitions in performance and loss-landscape metrics, and weight averaging.

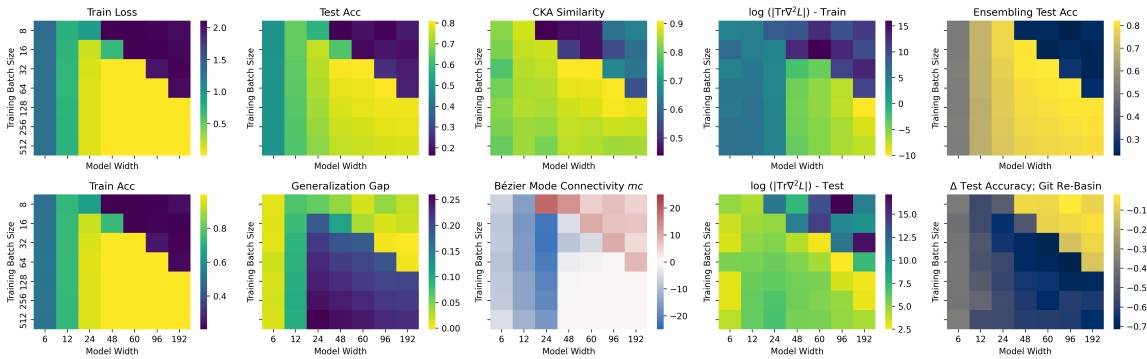

Figure 8: Phase plots for the CIFAR-10 ViT model zoo, showing distinct phase transitions in performance and loss-landscape metrics, ensembling, and weight averaging.

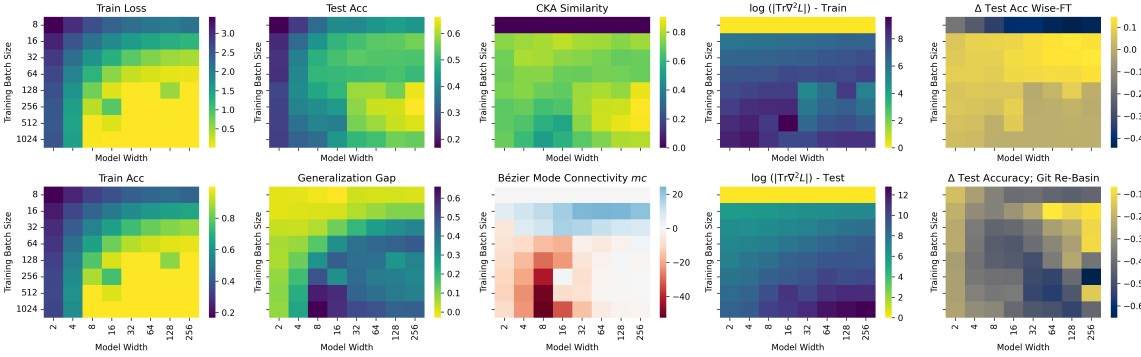

Figure 9: Phase plots for the CIFAR-100 ResNet-18 model zoo, showing distinct phase transitions in performance and loss-landscape metrics, and weight averaging.

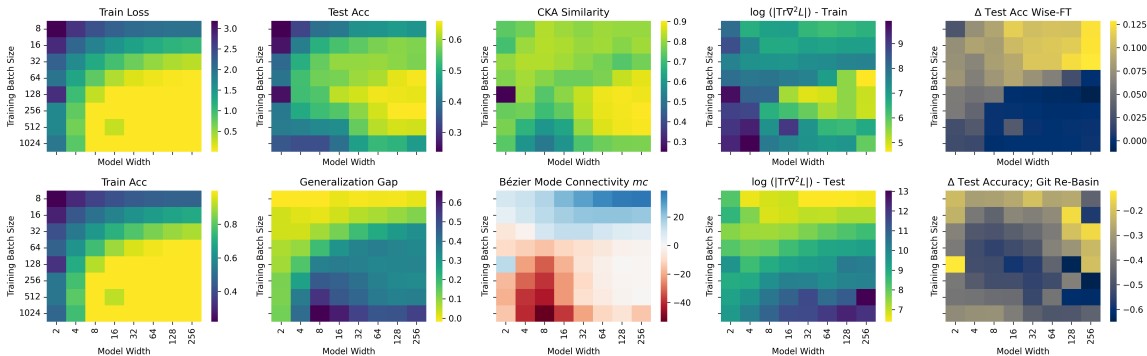

Figure 10: Phase plots for the CIFAR-100 ResNet-50 model zoo, showing distinct phase transitions performance, loss landscape metrics, and weight averaging.

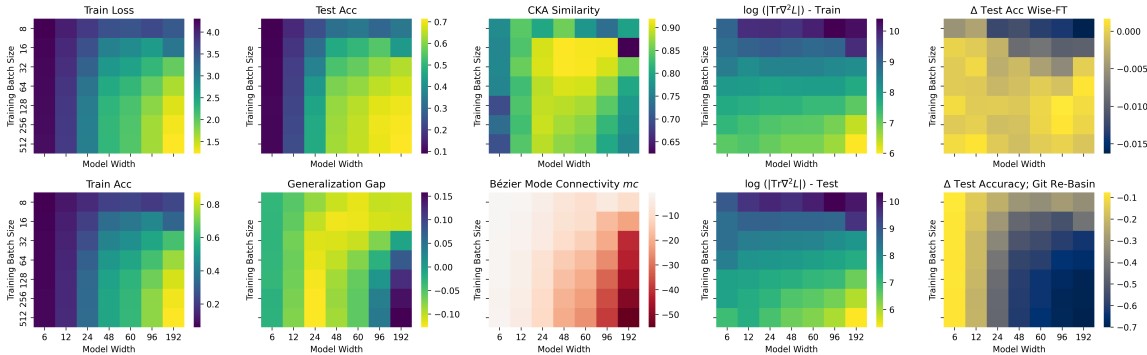

Figure 11: Phase plots for the CIFAR-100 ViT model zoo, showing distinct phase transitions in performance, loss landscape metrics, transfer learning and weight averaging.

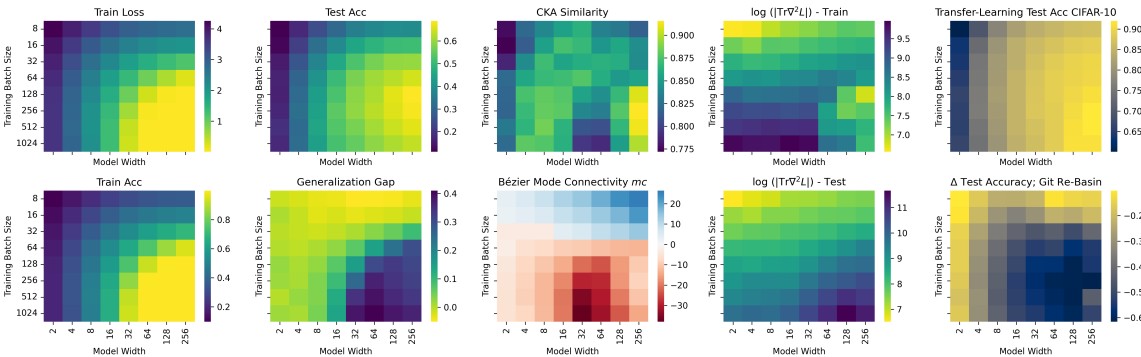

Figure 12: Phase plots for the Tiny-Imagenet ResNet-18 model zoo, showing distinct phase transitions in performance and loss-landscape metrics.

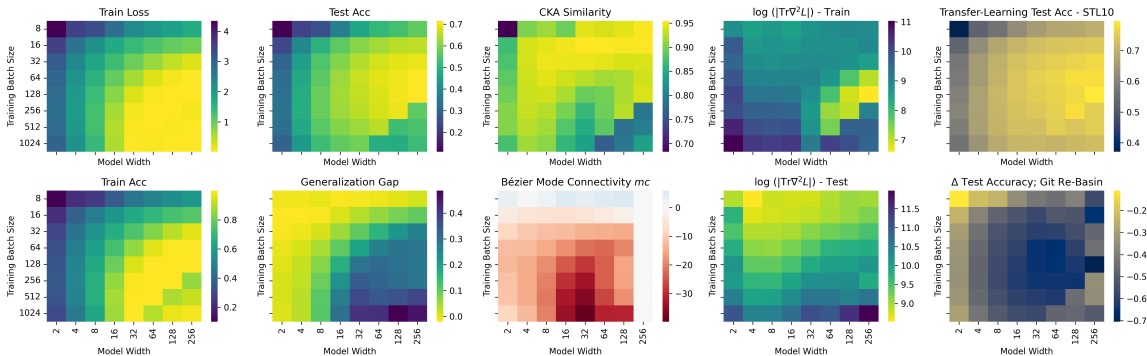

Figure 13: Phase plots for the Tiny-Imagenet ResNet-50 model zoo, showing distinct phase transitions in performance, loss landscape metrics, transfer learning and weight averaging.

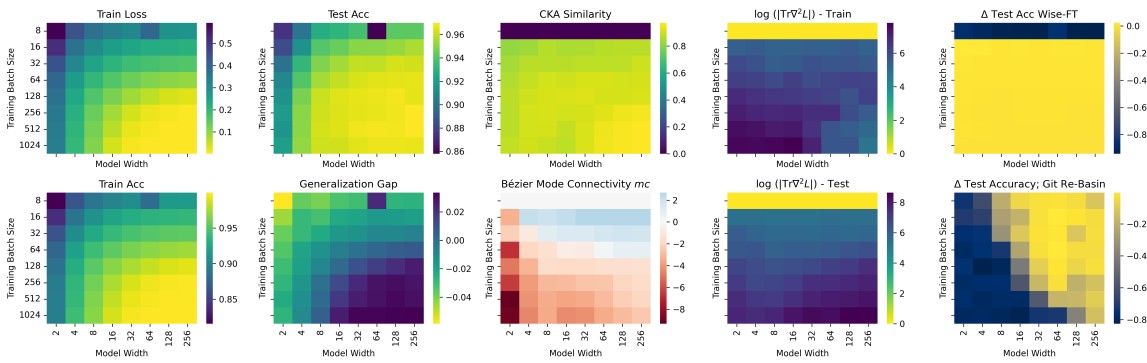

Figure 14: Phase plots for the SVHN ResNet-18 model zoo, showing distinct phase transitions in performance and loss-landscape metrics, and weight averaging..

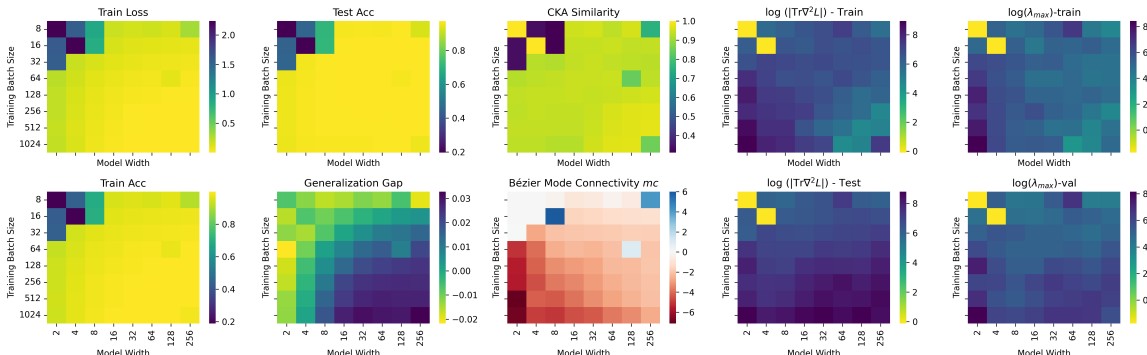

Figure 15: Phase plots for the SVHN ResNet-50 model zoo, showing distinct phase transitions in performance and loss-landscape metrics.

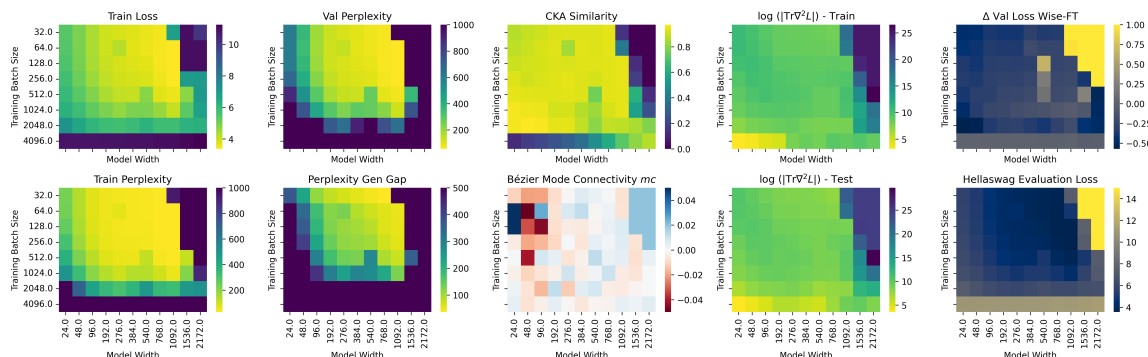

Figure 16: Phase plots for the openwebtext GPT-2 model zoo, showing distinct phase transitions in in performance and loss-landscape metrics, aligned with the vision model zoos. Please note that the reported metrics are adjusted to the language domain. Instead of accuracy, we report perplexity. Also, the bezier mode connectivity is computed on the loss, rather than on the error, which induces some noisiness.

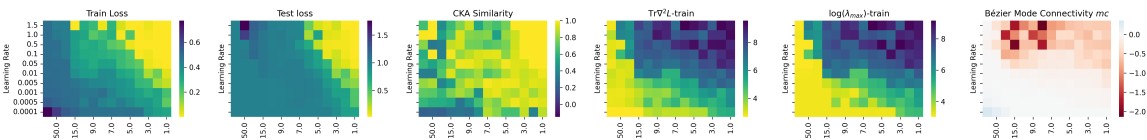

Figure 17: Phase plots for the 1D Convection PINN model zoo, showing distinct phase transitions in performance and loss-landscape metrics.

Table 4: Full list of hyperparameters of the vision model zoos. Width indicates the width of the first residual block. From that, we follow the same scaling factor as the standard ResNet.

| Base Architecture | ResNet-18, ResNet-50 | ViT |
|---|---|---|
| Datasets | SVHN, CIFAR10 CIFAR100, TinyImagenet | CIFAR10, CIFAR100 |
| Activation | ReLU | ReLU |
| Initialization | Kaiming Uniform | Kaiming Uniform |
| Optimizer | SGD | ADAMW |
| Learning Rate | 0.1 | $6e-3$ |
| Momentum | 0.9 | |
| WD | $5e-4$ | CIFAR10: $5e-4$. CIFAR100: $5e-2$ |
| LR Schedule | OneCycleLR with Cosine Annealing | OneCycleLR with Cosine Annealing |
| Width | 2, 4, 8, 16, 32, 64, 128, 256 | 6, 12, 24, 48, 60, 96, 192 |
| Batch Size | 8, 16, 32, 64, 128, 256, 512, 1024 | 8, 16, 32, 64, 128, 256, 512 |
| Seeds | 0, 1, 2 | 0, 1, 2 |

Table 5: Full list of hyperparameters of the lanugage model zoos. `model_dim` indicates the tokensize of the transformer blocks.

| Base Architecture | GPT-2 |
|---|---|
| Dataset | OpenWebText |
| Activation | GeLU |
| Initialization | Kaiming Uniform |
| Optimizer | ADAMW |
| Learning Rate | $6e-4$ |
| WD | $1e-1$ |
| LR Schedule | OneCycleLR with Cosine Annealing |
| `model_dim` | 24, 48, 96, 192, 276, 384, 540, 768, 1092, 1536, 2172 |
| Batch Size | 32, 64, 128, 256, 512, 1024, 2048, 4096 |
| Seeds | 1, 2, 3 |
| Block size | 1024 |
| Training steps | 350k |

Table 6: Full list of hyperparameters of the SciML model zoos.

| | |
|---|---|
| Base Architecture | MLP |
| Problem | 1D convection |
| Activation | Tanh |
| Initialization | Kaiming Uniform |
| Optimizer | LBFGS |
| Learning Rate | 0.0001, 0.0005, 0.001, 0.005, 0.01, 0.05, 0.1, 0.5, 1.0, 1.5 |
| Convection coefficient $\beta$ | 1, 2, 3, 4, 5, 6, 7, 8, 9, 10, 15, 30, 50, 70 |
| Line search function | Strong Wolfe |
| `model_dim` | (50,50,50,50,1) |
| Batch Size | Full batch |
| Seeds | 0, 123, 2023, 54321, 123456 |
| Max iteration | 100k |
| Histort size | 50 |
| Loss | MSE |
| Collocation points | 100 |

Table 7: Conventional Performance Metric Distribution of the Computer Vision Model Zoos.

| Arch. | Data | Train Loss $\mu \pm \sigma$ [min,max] | Test Loss $\mu \pm \sigma$ [min,max] | Train Acc $\mu \pm \sigma$ [min,max] | Test Acc $\mu \pm \sigma$ [min,max] | GGap $\mu \pm \sigma$ [min,max] |
|---|---|---|---|---|---|---|
| R-18 | SVHN | 0.10±0.11 [0.00,0.38] | 0.15±0.04 [0.11,0.27] | 0.97±0.03 [0.88,1.00] | 0.96±0.01 [0.92,0.97] | 0.01±0.02 [-0.04,0.03] |
| R-50 | SVHN | 0.06±0.07 [0.00,0.24] | 0.14±0.02 [0.11,0.18] | 0.98±0.02 [0.93,1.00] | 0.97±0.01 [0.95,0.97] | 0.01±0.02 [-0.02,0.03] |
| R-18 | C-10 | 0.08±0.19 [0.00,0.66] | 0.67±0.34 [0.32,1.98] | 0.97±0.06 [0.77,1.00] | 0.82±0.08 [0.65,0.91] | 0.16±0.07 [0.04,0.35] |
| R-50 | C-10 | 0.04±0.09 [0.00,0.52] | 0.60±0.30 [0.27,1.69] | 0.99±0.03 [0.82,1.00] | 0.84±0.07 [0.64,0.92] | 0.15±0.06 [0.05,0.33] |
| R-18 | C-100 | 0.45±0.79 [0.00,2.48] | 2.02±0.54 [1.24,3.89] | 0.88±0.21 [0.35,1.00] | 0.53±0.12 [0.29,0.69] | 0.35±0.15 [0.01,0.67] |
| R-50 | C-100 | 0.35±0.68 [0.00,4.61] | 1.78±0.55 [1.18,4.61] | 0.91±0.18 [0.01,1.00] | 0.57±0.11 [0.01,0.70] | 0.34±0.14 [-0.01,0.67] |
| R-18 | TI | 1.20±1.06 [0.01,3.42] | 1.91±0.48 [1.29,3.22] | 0.71±0.25 [0.23,1.00] | 0.55±0.12 [0.26,0.70] | 0.16±0.15 [0.03,0.41] |
| R-50 | TI | 1.05±0.96 [0.00,3.55] | 1.85±0.51 [1.21,3.63] | 0.74±0.22 [0.21,1.00] | 0.57±0.11 [0.22,0.72] | 0.17±0.15 [-0.02,0.49] |
| VIT | C-10 | 0.77±0.83 [0.00,2.18] | 1.72±0.45 [0.71,2.96] | 0.71±0.31 [0.17,1.00] | 0.59±0.23 [0.10,0.82] | 0.13±0.10 [-0.01,0.27] |
| VIT | C-100 | 2.96±0.94 [1.21,4.32] | 2.72±0.82 [1.74,4.15] | 0.37±0.24 [0.06,0.88] | 0.43±0.22 [0.09,0.72] | -0.05±0.07 [-0.13,0.16] |

Table 8: Performance Metric Distribution of the GPT-2 language model zoo. Performance metrics are computed on train and test splits of openwebtext.

| Train Loss $\mu \pm \sigma$ [min,max] | Test Loss $\mu \pm \sigma$ [min,max] | Train Perplexity $\mu \pm \sigma$ [min,max] | Test Perplexity $\mu \pm \sigma$ [min,max] |
|---|---|---|---|
| 0.91 ± 1.43 [0.08,10.05] | 5.97 ± 2.31 [3.86,11.29] | 152.29 ± 1765.46 [1.09,23065.60] | 9121.83 ± 21080.70 [54.49,79800.74] |

Table 9: Performance Metric Distribution of the SciML model zoo.

| Train Loss $\mu \pm \sigma$ [min,max] | Test Error $\mu \pm \sigma$ [min,max] |
|---|---|
| 0.31 ± 0.18 [0.00048, 0.81] | 0.81 ± 0.34 [0.021, 1.28] |

### A.4 Intended uses

The dataset is a repository of trained deep-learning models with phase transitions. It is mainly intended to study phase transitions on populations of neural network models. For every model, we include multiple checkpoints, representing different training epochs, to allow for the study of the training procedure. We also provide loss landscape metrics, to allow researchers to relate their findings with the structure of the loss landscape. The dataset is intended to allow researchers to **(i)** identify phases in different model properties or applications like the weight averaging examples in the main paper; and **(ii)** evaluate existing methods that rely on pre-trained models systematically on models of different phases, to get a better understanding under which conditions methods can be expected to perform well. Further examples of applications of our dataset are presented in our publication: model training, model property prediction, model generation, model combination, etc.

Please note that this dataset is intended for research on populations of models, not to further improve performance on specific tasks directly. The models in our zoo were selected for their diversity in phases, not optimized for performance on their specific datasets; there may exist generating factors combinations achieving better performance with similar architectures.

The dataset is entirely synthetic and does not contain personally identifiable information or offensive content. Authors bear all responsibility in case of violation of rights.

### A.5 Hosting, Licensing, and Maintenance Plan

The dataset is made publicly available and licensed under the Creative Commons Attribution 4.0 International license (CC-BY 4.0). It can be accessed at `https://phasetransitions.modelzoos.cc`, together with the corresponding code and instructions on how to use it.

The dataset is hosted by the University of St. Gallen and will be made available to the general public with detailed instructions and examples through the project's public GitHub repository. The University of St. Gallen will ensure maintenance and long-term availability. We further plan on extending the dataset towards more architectures, tasks, and domains, and invite the community to engage.

