# OpenReview forum: "A Model Zoo on Phase Transitions in Neural Networks"
_DMLR — Accepted by DMLR_

### Review · Reviewer_Jgpb · 2025-06-02

**Recommendation:** 3
**Confidence:** 2

**Summary Of Contributions:**

The submission releases the Phase Transition Model Zoos, which contains 12 zoos on three domains: computer vision (10 ResNet/ViT sets on SVHN, CIFAR-10/100, Tiny-ImageNet), natural language (GPT-2 on OpenWebText), and physical systems (physics-informed PINNs).

Every model is annotated with conventional metrics, as well as loss-landscape statistics (Hessian trace/eig-max, Bézier mode connectivity, and CKA similarity); multiple training-epoch checkpoints are included. The authors demonstrate that (i) phases emerge consistently across domains and architectures, (ii) phase boundaries propagate into downstream procedures such as fine-tuning, pruning, ensembling, and weight averaging (Fig. 5), and (iii) the dataset enables phase-aware evaluation in weight-space learning applications.

**Strengths:**

- Comprehensive models and annotations.
- Demonstrates the practical relevance of phases for fine-tuning, pruning, and weight averaging.

**Audience:**

Yes

**Claims And Evidence:**

The claim that 'models are deliberately trained to occupy the five phases' is not yet fully auditable, because the submission omits the concrete decision rule that converts continuous landscape metrics into discrete phase labels.

**Datasets And Benchmarks:**

Yes.

**Extended Submissions:**

N/A

**Limitations:**

See the weaknesses above.

**Requested Changes:**

- See the weaknesses above.
- Figures 3,4, and 5 are hard to follow due to the different colors. Improve colour choices
- There is an 'Appendix D' in Section 4.1. But there is no Appendix D.

**Strengths And Weaknesses:**

Strengths:
- Comprehensive models and annotations.
- Demonstrates the practical relevance of phases for fine-tuning, pruning, and weight averaging.

Weaknesses:
- The models are not representative enough. The Language Model Zoo is limited to GPT-2-nano and SciML Model Zoo, with a single 1-D PDE.
- The main point of the phases is not clear. There is no explicit decision rule for mapping a checkpoint to one of the five phases. Figures 3-4 visualise clusters, but a reader cannot reconstruct the numeric boundaries. The conclusions are qualitative, with limited statistical confidence intervals.

---

### Review · Reviewer_B4HL · 2025-07-15

**Recommendation:** 3
**Confidence:** 2

**Summary Of Contributions:**

Building on the temperature-load landscape proposed by Yang et al. (2021), the paper leverages the observation that models within the same phase exhibit homogeneous behaviors. This motivates the introduction of the temperature-load phase concept.

The paper demonstrates that temperature-load phases and their transitions consistently emerge across different domains, tasks, and model architectures.

The authors present a large-scale dataset characterizing weight space dynamics across diverse tasks, including NLP, computer vision, and scientific machine learning. The dataset spans multiple training epochs and phase regimes, and includes a range of loss surface metrics.

**Strengths:**

Aside from the strengths listed:
The paper is clear, although there are some typos.
The paper presents the first large-scale structured dataset for the weight space learning task.

**Audience:**

Yes

**Claims And Evidence:**

The claims are not well supported. It is listed in the weakness section.

**Datasets And Benchmarks:**

Yes, the paper provided such details.

**Extended Submissions:**

This paper was published as a workshop paper. It does meet the eligibility criteria.

**Limitations:**

Similar to the weakness:

1. The phase separation is currently defined visually or heuristically, without formal quantitative boundaries or metrics. This limits reproducibility and may hinder broader adoption of the temperature-load phase framework.

2. The paper highlights interesting empirical patterns (e.g., CKA values, ruggedness of loss surfaces), but does not attempt to explain or theoretically interpret them. This may limit the scientific insight derived from the dataset.

3. Despite the dataset's scale, its practical utility is demonstrated only through a single pilot experiment using a linear probe. This restricts the strength of claims regarding its general applicability and usefulness.

4. Although the dataset is proposed as a foundation for phase-aware training strategies, no empirical validation or case study is provided to support this use case.

5. If the criteria for phase assignment and metrics are not released or explained, it may be challenging for future researchers to replicate the phase identification process or extend the dataset.

**Requested Changes:**

To clarify the weakness in terms of the requested changes:

Critical to Address for Acceptance
1. Clarify Phase Separation and Related Work
The method for identifying and separating phases remains unclear. Are there quantitative metrics or thresholds used to define the phases in Figures 2 and 3? Introducing formal criteria and grounding the approach in existing literature on phase transitions would improve clarity and rigor.

2. Insufficient Analysis of Behavioral Homogeneity Within Phases
The paper claims that models within the same phase exhibit similar behavior, yet offers no substantial analysis to support this. A detailed exploration of whether phases correspond to consistent loss landscape properties or optimization behavior is necessary to validate the central hypothesis.

3. Lack of Comprehensive Evaluation of the Dataset's Utility
The only evaluation provided is a minimal linear-probe experiment. To establish the value of the dataset, the authors should include additional experiments. This is essential to demonstrate that the dataset is indeed more informative than previous WSL datasets.

4. Demonstrate Relevance to Phase-Aware Training Algorithms
Although the dataset is positioned as a resource for developing and evaluating phase-aware training methods, the paper provides no such demonstration. A toy example or a brief case study would substantiate this claim.

Suggestions

1. Explanation of Visualizations in Figure 2
The subfigures in Figure 2 are interesting but are not well-explained. For instance, the rugged loss surfaces in Phases I and II, or the unexpected CKA patterns (low in II and IV-B but high in IV-A), are left unexplained. Providing intuition or hypotheses would enhance interpretability.

**Strengths And Weaknesses:**

Strengths:

1. The paper presents compelling empirical evidence that temperature-load phases and their transitions consistently appear across a wide range of domains, tasks, and model architectures.

2. It introduces a large-scale, well-structured zoo of trained models spanning diverse tasks (NLP, CV, SciML), training epochs, and phases, which is a valuable resource for future research on weight space learning.

3. The authors validate the potential utility of their dataset through a pilot experiment, demonstrating that simple statistics of model weights can predict test accuracy and loss surface metrics.

Weaknesses:

1. The definition and separation of the phases are not rigorously established. The paper lacks a concrete connection to related work on phase transitions in learning dynamics, and it remains unclear whether there are formal metrics or algorithms to quantitatively delineate the phase boundaries, particularly in Figures 2 and 3.

2. Figure 2 lacks sufficient explanation. For instance, the rugged basins observed in Phases I and II are not discussed, nor is the behavior of CKA values across phases clearly explained. The finding that CKA is small in Phases II and IV-B but large in Phase IV-A is counterintuitive and warrants clarification, as it could provide meaningful insights into phase-related properties.

3. While the dataset is positioned as a tool for evaluating phase-aware training algorithms, the paper does not provide evidence or experiments demonstrating its effectiveness in that context. Including such an application would significantly strengthen the paper’s impact.

4. Although the central motivation is that models within the same phase exhibit similar behaviors, the paper does not thoroughly explain or explore this direction. A deeper analysis of how phases relate to loss surface metrics or optimization dynamics would add significant value.

5. Despite the dataset’s scale and coverage, the only evaluation provided is a small-scale linear probe-based pilot study. Additional experiments, such as comparisons with other weight space datasets, would better illustrate the dataset’s effectiveness and improve accessibility for the broader community.

---

### Review · Reviewer_Udhr · 2025-07-25

**Recommendation:** 3
**Confidence:** 2

**Summary Of Contributions:**

This paper presents a comprehensive evaluation of neural network model zoos designed to systematically cover phase transitions in the loss landscape. The authors create 12 model zoos spanning computer vision, natural language processing, and scientific machine learning domains. In the proposed zoo, each model is annotated with performance metrics and loss landscape properties including Hessian-based metrics, mode connectivity, CKA similarity to validate coverage of five distinct phases identified in prior work. The authors demonstrate that phase transitions manifest across different architectures and domains.

**Strengths:**

See feedback in Strengths.

**Audience:**

Yes

**Broader Impact Concerns:**

No significant ethical concern found.

**Claims And Evidence:**

The claims made in the submissionare generally supported by accurate, convincing and clear evidence.

**Datasets And Benchmarks:**

The full dataset access is not provided (only samples are accessible) — the authors should provide full access to satisfy the DMLR requirements before acceptance. Other requirements are satisfied.

**Extended Submissions:**

Eligibility criteria satisified. This paper is only published in ICLR 2025 Workshop: Neural Network Weights as a New Data Modality.

**Limitations:**

See feedback in Weaknesses.

**Requested Changes:**

1 - I suggest the authors to offer concrete strategies for reliably achieving specific phases during training or selecting models based on phase characteristics for particular applications (see weaknesses 1).

2 - An in-depth understanding of what causes phase transitions will be beneficial to the researchers who do not work in this domain, and could benefit researchers in a broader community. Hence I suggest to add additional investigation into what causes phase transitions (see weaknesses 2).

3 - I suggest to revise the evaluation protocol in domains other than cv and enrich the experiments (see weaknesses 3).

4 - I suggest to add justification about why these particular metrics are optimal or compare them against alternative measures that might provide different insights into the phase structures (see weaknesses 4).

**Strengths And Weaknesses:**

Strengths:

1 - The authors provide a well-structured dataset that varies load-like and temperature-like parameters across multiple domains and architectures. The inclusion of detailed loss landscape metrics for each model represents a key contribution to the weight space learning community.

2 - The demonstration that phase transitions generalize across computer vision (ResNet, ViT), natural language processing (GPT-2), and scientific machine learning (PINNs) shows compelling evidence for the universality of these phenomena.

3 - The authors show how phase information influences real-world applications like fine-tuning, pruning, and model averaging. The systematic evaluation framework has the potential to enable researchers to understand when and why certain methods succeed or fail.

Weaknesses:

1 - While the paper successfully demonstrates that phases exist across domains and affect downstream performance, it provides limited actionable guidance for practitioners. The paper will benefit from offering concrete strategies for reliably achieving specific phases during training or selecting models based on phase characteristics for particular applications.

2 - The paper lacks deep investigation into what causes phase transitions. The temperature-load parameterization appears a limited abstract of the underlying mechanisms, which could have provided better theoretical understanding.

3 -  Despite authors' claim of generalizability, the dataset is heavily skewed toward cv classification tasks. The language model zoo uses only GPT-2 variants, and the scientific ML component covers only physics-informed neural networks on a single PDE. These limited evaluation protocol can lead to biased observations from domains other than cv, and the paper will benefit from enriching the scope of domains other than cv.

4 - The choice of specific loss landscape metrics (Hessian trace, mode connectivity, CKA similarity) may not capture all relevant aspects of phase transitions. I suggest to add justification about why these particular metrics are optimal or compare them against alternative measures that might provide different insights into the phase structures.